# Mucins: the Old, the New and the Promising Factors in Hepatobiliary Carcinogenesis

**DOI:** 10.3390/ijms20061288

**Published:** 2019-03-14

**Authors:** Aldona Kasprzak, Agnieszka Adamek

**Affiliations:** 1Department of Histology and Embryology, Poznan University of Medical Sciences, Swiecicki Street 6, 60-781 Poznań, Poland; 2Department of Infectious Diseases, Hepatology and Acquired Immunodeficiencies, University of Medical Sciences, Szwajcarska Street 3, 61-285 Poznań, Poland; agnieszkaadamek@ump.edu.pl

**Keywords:** primary liver cancer, mucins, liver cancer immunophenotype, tissue expression, mucins as oncogenes, hepatobiliary carcinogenesis

## Abstract

Mucins are large *O*-glycoproteins with high carbohydrate content and marked diversity in both the apoprotein and the oligosaccharide moieties. All three mucin types, trans-membrane (e.g., MUC1, MUC4, MUC16), secreted (gel-forming) (e.g., MUC2, MUC5AC, MUC6) and soluble (non-gel-forming) (e.g., MUC7, MUC8, MUC9, MUC20), are critical in maintaining cellular functions, particularly those of epithelial surfaces. Their aberrant expression and/or altered subcellular localization is a factor of tumour growth and apoptosis induced by oxidative stress and several anti-cancer agents. Abnormal expression of mucins was observed in human carcinomas that arise in various gastrointestinal organs. It was widely believed that hepatocellular carcinoma (HCC) does not produce mucins, whereas cholangiocarcinoma (CC) or combined HCC-CC may produce these glycoproteins. However, a growing number of reports shows that mucins can be produced by HCC cells that do not exhibit or are yet to undergo, morphological differentiation to biliary phenotypes. Evaluation of mucin expression levels in precursors and early lesions of CC, as well as other types of primary liver cancer (PLC), conducted in in vitro and in vivo models, allowed to discover the mechanisms of their action, as well as their participation in the most important signalling pathways of liver cystogenesis and carcinogenesis. Analysis of mucin expression in PLC has both basic research and clinical value. Mucins may act as oncogenes and tumour-promoting (e.g., MUC1, MUC13), and/or tumour-suppressing factors (e.g., MUC15). Given their role in promoting PLC progression, both classic (MUC1, MUC2, MUC4, MUC5AC, MUC6) and currently tested mucins (e.g., MUC13, MUC15, MUC16) have been proposed to be important diagnostic and prognostic markers. The purpose of this review was to summarize and update the role of classic and currently tested mucins in pathogenesis of PLC, with explaining the mechanisms of their action in HCC carcinogenesis. It also focuses on determination of the diagnostic and prognostic role of these glycoproteins in PLC, especially focusing on HCC, CC and other hepatic tumours with- and without biliary differentiation.

## 1. Introduction

Mucins (MUC) are a major constituent of any mucous secretion, providing the mucus with its biophysiochemical properties due to their nature and extent of glycosylation [1,2]. They are mostly responsible for formation of the protective barrier of the mucous membranes [1,2,3,4,5]. Additionally, they serve many more specialized functions, such as regulation of solute transport or provision of attachment sites for commensal and pathogenic microbes, as well as for leukocyte targeting [6,7,8]. They are associated with cellular regeneration, differentiation, integration, signalling, adhesion and apoptosis [3,9]. Despite their first purified fraction being isolated from the mucus of the cervical epithelium [10], they were most commonly described in the epithelia of the intestine/colon and airways [2,7,8].

Mucins are large O-glycoproteins with high carbohydrate content and marked diversity both in the apoprotein and in the oligosaccharide moieties [5]. They are produced by a variety of secretory epithelial cells (including endothelial cells) and leukocytes [4,5,7]. The main mucin producers are goblet cells and mucous cells in surface epithelium and glands of the gastrointestinal (GI) tract [5,8,11,12].

Human mucin genes exhibit a specific domain, called the Variable Number Tandem Repeat Region (VNTR). It encodes a Tandem Repeat Peptide (TRP) with high percentage of such amino acids (aa) as serine (Ser) and threonine (Thr). Finally, they exhibit complex RNA expression patterns [3,13]. These genes code glycoproteins of large molecular mass (in GI tract from 250 to 2000 kDa and from 10,000 to 30,000 kDa in the respiratory system), in which the carbohydrates are bound with, among others, proline (Pro), Ser and Thr (PTS domain), through O-glycosidic bonds [14,15,16].

The protein fragment of a mucin (apomucin) constitutes to ~20% of their dry mass [15]. Carbohydrates make up 70–80% of the molecular mass (mostly ~50%) [11,13,14,17]. Carbohydrate chains are present in the VNTR regions of the aa, with their size and number depending on mucin type [17]. Tight carbohydrate packing along the polypeptide chain is responsible for the filament structure of the mucins, critical for the function they serve. The attached sulphate and sialic moieties are responsible for the strong negative charge on the mucin surface [14]. Presence of Lewis antigens is a modification of oligosaccharide chains [18,19,20,21].

21 mucin genes were identified in human, among which 15 are expressed in different regions of the GI system [22]. Based on the structure, function and cellular localization, the mucins are divided into two classes: membrane (cell surface associated) and secreted [11,16,23,24]. This division also comes from an observation that some of the integral membrane proteins present on the apical part of different epithelial cells, as well as leukocytes, are also classified as mucins [6]. Membrane-bound mucins are modular proteins with structural organization usually containing PTS, Epidermal Growth Factor (EGF)-like and Sea Urchin Sperm Protein, Enterokinase and Agrin (SEA) domains [11,15,25]. Via these domains, membrane-bound mucins are involved in cellular adhesion, pathogen binding and signal transduction [15], while secreted mucins are highly related to the viscoelastic properties of mucus [24]. Sometimes, three subfamilies of mucins are described: membrane-bound/trans-membrane (e.g., MUC1, MUC4, MUC16), secreted (gel-forming) (e.g., MUC2, MUC5AC, MUC6) and soluble (non-gel-forming) (e.g., MUC7, MUC8, MUC9, MUC20) [3]. It is worth noting that MUC1, MUC4 and MUC16 evolved from distinct ancestors and that the trans-membrane mucins consist of different subgroups based on their genetic backgrounds. Additionally, it was shown that the MUC1 SEA domain originates from heparin sulphate proteoglycan of basement membrane (HSPG2; perlecan), an inducer of tumour cell growth [26].

Mucins serve a role in both physiological and pathological conditions [3,16,20,21,27,28,29]. Aberrant expression of mucins can contribute to loss of epithelial cell polarity and promote Epithelial-Mesenchymal Transition (EMT), which leads to enhanced cell motility and invasion ability, a key step for tumorigenesis [16,30,31]. During neoplastic transformation of the GI tract, mucins are considered as diagnostic-prognostic markers [14,29,32,33], as well as therapeutic targets [16,34,35]. Many pathways that involve mucins were described in the carcinogenesis process, involving Ras, β-catenin, p120 catenin, p53 and oestrogen receptor α, Hepatocyte Growth Factor/tyrosine-protein kinase Met (HGF/c-Met), Nuclear Factor Kappa-light-chain-enhancer of Activated B cells (NF-κB), Mitogen-activated Protein Kinase (MAPK), insulin, Transforming Growth Factor β (TGF-β), Vascular Endothelial Growth Factor (VEGF) and c-Jun N-terminal kinase/Activation Protein 1 (c-JNK/AP-1) [36,37,38,39,40,41,42,43,44].

Altered expression profiles of mucins, as well as quantitative (up and down regulation) and qualitative (e.g., disturbances of glycosylation) changes in apomucin and structures of O-glycans, are often seen in GI tract tumours, with these alterations greatly contributing to the phenotype and biology of cancer cells [14,20,21,29].

It was widely believed that hepatocellular carcinoma (HCC) does not produce mucins, whereas cholangiocarcinoma (CC) or combined/mixed hepatocellular-cholangiocarcinoma (cHCC-CC) may produce these glycoproteins [45,46,47,48,49,50,51]. However, a growing number of reports shows that mucins can be produced by HCC cells that do not exhibit or are yet to undergo, morphological differentiation to biliary phenotypes [38,52,53,54,55,56]. The detailed role of tissue expression of mucins in neoplastic cells of HCC in vivo have been poorly recognized.

The purpose of this review was to summarize and update the role of classic (MUC1, MUC2, MUC4, MUC5AC, MUC6) and currently tested (e.g., MUC13, MUC15, MUC16) mucins in pathogenesis of PLC, with explaining the mechanisms of their action in HCC carcinogenesis. It also focuses on determination of the diagnostic and prognostic role of these glycoproteins in PLC, especially focusing on HCC, CC and other hepatic tumours with- and without biliary differentiation.

## 2. Short Overview of Gastrointestinal (GI) System Mucins

Several excellent reviews describe structure and biochemical characteristics of mucins present in the GI tract [11,14,57]. According to modern proteomics, MUC7 and MUC19 are main components of the saliva [11,58], MUC5AC and MUC6 are characteristic for the stomach mucus, with MUC2 being proprietary to intestinal mucus [7,8,11,57].

### 2.1. Secreted Mucins

Mucins of this group are coded by the mucin gene family, including: *MUC2, MUC5AC, MUC5B, MUC6* (all localised on the 11p15.5 chromosome) and *MUC19* (12q12) as oligomerizing mucins and *MUC7* (4q13.3) as the only non-oligomerizing mucin [14,22]. They create an impermeable gel that forms a physical barrier and “trap” microorganisms [59]. The *MUC7* gene encodes the main saliva protein—MUC7 (also called monomeric mucin, MG2), produced by salivary glands. *MUC5B* gene encodes a salivary mucin, also called MG1 [11]. Both those mucins take part in the processes of agglutination and bacterial colonization on the surfaces of the oral cavity. MUC19 is also present in the saliva [58].

The MUC5A (gastric foveolar mucin) is expressed in the gastric surface epithelium, while MUC6 (pyloric gland-type mucin) is produced by the cells of gastric glands [11,14]. MUC2 (goblet cell mucin) is the main mucin expressed by goblet cells in small and large intestine [8,11,59]. What is also interesting, the surface colonic goblet cells express mucins continuously to maintain the inner mucus layer, whereas goblet cells of the small intestinal and colonic crypts only secrete upon stimulation, for example, after endocytosis or in response to acetyl choline [8]. MUC5B is present in lower amounts and is produced by a fraction of the goblet cells present at the bottom of the colonic crypts in humans [14].

### 2.2. Cell Surface (Trans-Membrane) Mucins

In GI system mucins of that subfamily are coded by nine genes localised on different chromosomes. These genes are: *MUC1* (chromosome location: 1q22), *MUC3A* (7q22.1), *MUC3B* (7q22), *MUC4* (3q29), *MUC12* (7q22.1), *MUC13* (3q21.2), *MUC15* (11p14.3), *MUC16* (19p13.2) and *MUC17* (7q22.1) [14,22]. Proportions in their expression are dependent on the location in the GI tract, for example MUC1 (“polymorphic” epithelial mucin) is mostly present in the stomach and the pancreatic epithelium, MUC3 can be found in the intestine, while MUC4 (“tracheobronchial” mucin) is characteristic for the oral cavity and intestine [14]. Many regions of the GI tract are able to produce different mucin types, with a possibility of co-expression of more than one type of mucins by singular cells [14,28]. All of the mucins of this group are expressed on the apical part of the epithelial cell membrane and usually exhibit large extracellular domains. Additionally, it is worth noting that intestinal trans-membrane mucins show stronger expression in enterocytes, compared to goblet cells [11,14]. MUC3, MUC12 and MUC17 probably build the glycocalyx [11]. General mechanisms of trans-membrane mucin action, especially those of MUC1 [so called Cancer Antigen 15-3 (CA15-3) or CD227], MUC4 and MUC16 (CA125), as well as their role in tumorigenesis and metastasis are relatively well known [22,30,36,43,60].

MUC1 is the best characterized of those proteins, consisting of two subunits: C- and N-terminal [31]. MUC1 participates in many cellular signal transduction pathways. It can create complexes with numerous transcription factors inside the nucleus, as well as cooperate with mitochondrial proteins [11,36,60]. MUC1 C-terminal subunit (MUC1-C) signals drive the processes of integration of the EMT induction with activation of stem cell traits, epigenetic reprogramming and immune evasion [31]. Stimulation of pluripotent stem cell growth through a membrane-anchored MUC1 cleavage product, so-called MUC1*, was also described as a major mechanism present in neoplastic cells [61].

MUC4 is an intramembranous ligand for erythroblastic oncogene B2 (ErbB2) receptor tyrosine–protein kinase, related to regulation of p27, which is a cyclin-dependent kinase inhibitor involved in the control of G1 and S phases of the cell cycle. MUC4/sialomucin complex acts as a regulator of differentiation and a modulator of cell proliferation when functioning synergistically with neuregulin [62].

MUC15 is a highly glycosylated protein found in bovine milk fat globule membranes, as well as in small intestine, colon and foetal liver [63].

MUC16 (CA125)—the largest glycoprotein (3–5 million Da) in the Mucins family [5], has been identified as a prominent cancer biomarker in vivo, especially for epithelial ovarian cancers [43], with immunohistochemical (IHC) expression also reported in patients with liver diseases [64].

Most of the studies on the role of mucins in carcinogenesis concerns human breast cancer, colorectal carcinoma, pancreatic and ovarian cancers in in vitro and in vivo models [30,36,43,60].

Research on the mechanisms of mucin action in PLC carcinogenesis are also conducted [38,42,44,52,53,65] and will be further described in the following chapters of this manuscript.

## 3. Mucins Expression in Normal Liver

Mucin production was described in the biliary tract (BT) and associated glands (liver, pancreas) in prenatal [66,67,68,69,70,71,72,73], as well as in postnatal biliary tract development [74,75,76].

### 3.1. Mucins and Biliary Tract Development

The human BT is formed from the hepatic diverticulum, a structure which develops from the embryonic foregut in the 4th week of gestation (WG). The caudal part of the hepatic diverticulum is modelled from the mesenchyme to form the gallbladder, cystic duct and extrahepatic bile duct (EBD), while the rostral portion gives rise to the liver and the intrahepatic biliary system [77]. During liver development, hepatocytes and biliary epithelial cells (cholangiocytes) each arise from a common progenitor (hepatoblast) [78]. The human EBD is a well-defined tubular structure by the 6th WG, whereas the intrahepatic biliary system during this period of gestation is represented by the primitive ductal plate (DP) [70,72,77]. The distal portions of the right and left hepatic ducts develop from the EBDs and form clearly defined tubular structures by 12 WG [70].

Cholangiocytes of EBD showed presence of neutral and acidic mucins at 23–40 WG. Extrahepatic peribiliary glands (EPBGs) (emerged from EBDs around 36 WG) also showed strong expression of neutral and acidic mucins. Apomucin differentiation was also detected in EBD development, through identification of MUC1 and MUC6 expression in cholangiocytes of EBD and EPG [72].

The human intrahepatic biliary ducts (IBDs) are organized into large, septal, interlobular bile ducts and bile ductules. Small (S) cells concern both intralobular bile ductular cells and small isolated biliary cells scattered in hepatic parenchyma [74]. IBDs develop from DP, a double-layered cylindrical structure located at the interface between hepatoblasts and portal mesenchyme in human foetal livers. The following stages of IBD development include: DP remodelling, remodelled DP and mature IBDs [73,77]. Some of the authors did not manage to detect expression of mucin mRNA (MUC1, MUC2, MUC3, MUC4, MUC5AC, MUC5B, MUC6 and MUC7) at any gestational age [69]. Others confirmed lack of MUC2 expression throughout the foetal IBD development but described MUC1 expression in the all four of its stages. MUC5AC and MUC6 were only present in the DP [71,73]. Carboxylated and sulphated mucin residues were only detected in mature IBD in foetal livers, while the DP frequently showed neutral mucins, with much less frequent acidic mucin occurrence [71]. Similarly, Sasaki et al. described common expression of MUC1 in new bile ducts of the portal tracts, either at the hilar (corresponding to the large IBDs) or peripheral level (corresponding to the small IBDs), as well as focal MUC1 expression in DP [79].

The intrahepatic PBGs arise at 7 WG from the immature periportal primordial hepatocytes at the hepatic hilum and are located around the large IBD. The differentiation of PBGs into mucus acini and ectopic exocrine pancreatic tissue occurs around 3 months after birth [66,67,79]. PBGs are histologically divided into intramural and extramural structures. Intramural PBGs consist of simple tubular glands with large mucin expression, sparsely and irregularly distributed within the ductal wall. The extramural PBGs are characterized by the presence of excretory units that consist of seromucinous acini and a conducting system in the periductal tissue. Both PBGs secrete neutral and acid mucins into the ductal lumen. The PBGs appear in the late foetal period and complete their development about 15 years after birth [67].

In embryonic liver, mRNA of MUC3 was already expressed at 6.5 WG in the perinuclear region of hepatoblasts. This pattern of expression remained constant until 18 WG. Later, the labelling was weak and distributed throughout the cytoplasm. MUC1, MUC2, MUC4, MUC5AC, MUC5B, MUC6 and MUC7 were not detected in hepatoblasts or primordial hepatocytes at any gestational age [69].

### 3.2. Mucins and Postnatal Liver

The main mucin producing cell type in postnatal liver are cholangiocytes of IBDs, EBDs and PBGs [38,68,69,74]. The main mucin genes expressed in cholangiocytes were *MUC3*, *MUC6* and *MUC5B*. Only weak expression of *MUC1*, *MUC5AC* and *MUC2* and no expression of *MUC4* and *MUC7* was observed [68]. Similarly, highly glycosylated MUC1 was rare and focally expressed in cholangiocytes of small IBDs [80,81]. However, other authors, using anti-KL-6 mucin (sialylated and hyperglycosylated form of MUC1), showed the expression of this protein on the apical surface of normal bile duct cells [32]. Other study showed that cholangiocytes of large IBDs constantly expressed MUC3, whereas those of small IBDs did not. MUC2 and MUC5/6 were absent in the postnatal livers [79]. Later study of Sasaki et al. showed weak focal cytoplasmic expression of MUC6 in septal IBDs, with no (or only focal) expression of this mucin in interlobular IBDs, bile ductules and/or S cells [74]. Apart from cholangiocytes, MUC3 was also found in hepatocytes [68]. No expression of MUC1/KL-6 mucin was detected in normal hepatic parenchyma (including hepatocytes) [32,38,82,83].

## 4. Role of Mucins in Hepatobiliary Carcinogenesis

Role of mucins in hepatobiliary epithelial carcinogenesis has been previously suggested and mainly concerns intraductal papillary neoplasm of the biliary ducts (IPNB) with glandular involvement [47,50,75,84,85,86,87,88,89,90,91]. Recently, a growing number of works state that mucins can also be produced by HCC cells that do not exhibit or are yet to undergo, morphological differentiation to biliary phenotypes [38,52,53,54,55,56].

Primary liver cancer comprises HCC, intrahepatic CC (iCC) and other rare tumours, notably cHCC-CC, fibrolamellar carcinoma and paediatric hepatoblastoma. PLC is the second leading cause of cancer-related death worldwide [78,92]. The HCC (~90% of all PLC) and iCC have been described as independent tumours that originate from distinct cell populations, as well as exhibit distinct histological, molecular and clinical features but share some of the risk factors and oncogenic pathways [78,92,93,94]. The iCC exhibits predominant intraductal papillary growth in the bile ducts, known as “papillary” and “intraductal” types of BT carcinoma (BTC) [88,95,96]. Combined HCC-CC is an uncommon form of PLC, having features of both hepatocellular and biliary epithelial differentiation, with occurrence of 0.4%–14.2% of PLC cases [97]. It is either classified into the “classical” subtype or a subtype exhibiting “stem-cell features” [98,99].

Numerous risk factors have been hypothesized to be involved in hepatobiliary carcinogenesis [93,94,95,96]. Liver cirrhosis, with or without HBV/HCV-associated chronic infection, is a common risk factor in HCC and iCC, whereas PSC, biliary duct cysts, hepatolithiasis and parasitic biliary infestation with flukes are more often noted in iCC [94].

### 4.1. Cell Origin of Mucin-Producing Primary Liver Cancer

Several cell types have the longevity and self-renewal ability required to become the cell of PLC origin. Among them, hepatocytes, cholangiocytes and hepatic stem/progenitor cells (HPCs) can be distinguished. Two HPC niches, canals of Hering and intrahepatic PBGs, have been described within the liver [78,100,101,102,103]. HPCs (or oval cells) are depicted as bipotential progenitor cells that can give rise to both hepatocytes and cholangiocytes [78]. Recently, it has been reported that adult hepatocytes can be a direct origin of HCC, due viral infection (e.g., HBV, HCV) induced transformation and sequential genomic insults. Large hepatocyte plasticity favours dedifferentiation into hepatocyte precursor cells that express progenitor cell markers or transdifferentiation into biliary-like cells, which give rise to iCC. HPCs also give rise to HCC and iCC with progenitor-like features, whereas adult cholangiocytes, can only give rise to iCC [78].

In practice, HCC most commonly originates from regenerating liver cells with genetic alterations in liver chronically inflamed due to HBV, and/or HCV. Recently, Matsumoto et al. demonstrated that proliferating ductal cells (PDCs) expressing EpCAM, one of cancer stem cell markers, also give rise to HCC in inflamed liver [104]. This confirms the hypothesis of stem/progenitor-derived hepatocarcinogenesis. There are also some results that indicate the involvement of similar long-term disease processes in the development of viral hepatitis-associated CC and viral hepatitis-associated HCC. Both viral hepatitis-associated liver diseases shared common carcinogenesis process and both possibly arose from the HPCs [105].

It is still discussed which cells are the precursors of the mucin-producing PLC. It is suggested that intrahepatic PBGs (containing multipotent bile tree stem cells) could be implicated in carcinogenesis of mucin-producing iCC [91,96,102,103,106]. Mucin expression was also identified in many cells of EPBGs, which contain cells of mature and immature phenotypes and proliferate in response to bile duct injury (e.g., induced by virus infection) [106].

It is said that mucin-producing-iCCs have similar origin to hilar and extrahepatic CCs, as they exhibit numerous morphological, clinical, immunohistochemical and molecular similarities with this CC subtypes [95,96,102,107]. It has been proven that these CCs can arise from cylindrical mucin-producing cholangiocytes located in large bile ducts. In the same time, mixed-iCCs had a profile similar to that of cholangiolocellular carcinoma (CoCCs) [107]. The cellular origin of these tumours is usually associated with HSCs of canals of Herring [108]. CoCCs are very rare malignant tumours (1% of all PLCs), previously classified as a subtype of iCC [92] or cHCC-CC [98]. 100% of CoCCs exhibit hepatocellular differentiation in parts of the tumours and typical expression of HPC markers [108]. In turn, study of Maeno et al. suggests that CoCCs may originate from interlobular ducts of liver rather than from cholangioles. The positive rate of MUC1 (~7%) in CoCCs was significantly lower than in any control group (~18%–30%) [109]. Recent studies showed colocalization of MECA-79 (a cholangiocyte marker) and MUC1 on apical membrane of cholangiocytes with cholangial/ductular differentiation. Hence, CoCCs are also regarded as a subtype of peripheral-type iCC [110]. On the other hand, the most recent molecular research indicates that CoCCs are distinct biliary-derived entities associated with chromosomal stability and active TGF-β signalling, without any traits of the HCCs [111].

### 4.2. Mucins as a Potential Oncogenic Factors in Primary Liver Cancer

Among all mucins, MUC1 the foremost is proposed as an “oncofoetal” antigen in the hepatobiliary system [81] and plays key role as an oncogene in human hepatic tumorigenesis [38,39,40,41,42,44,112].

The most commonly described signalling pathways regulated by MUC1 include the Wnt/β-catenin [38,39], HGF/c-Met [38], TGF-β [39,40,44], MAPK with JNK/AP-1 [39,42] and JAK2/STAT3 signalling pathways [113]. Additionally, functional correlations between MUC1 and NF-κB [39], insulin [39] and VEGF pathway components were observed [39]. It was shown that MUC1-mediated protection against irradiation-induced apoptosis is associated with activation of the JAK2/STAT3 signalling pathway and induction of anti-apoptotic proteins Mcl-1 and Bcl-xL [113]. Other studies have revealed that MUC1 cytoplasmic tail can bind directly to pro-apoptotic proteins Bax or caspase-8 and regulate growth in HCC cells (SMMC-7721 cells) through Bax-mediated mitochondrial and caspase-8-mediated death receptor apoptotic pathways [114]. Mediation of various signalling pathways by MUC1 results in the occurrence basic processes associated with cancer (e.g., increase in cell proliferation, migration/invasion, survival; inhibition of apoptosis), as well as the whole tumour (e.g., increase in tumour growth; tumour metastasis, enabled by EMT, etc.) (Figure 1).

Studies by Xu et al. showed that after treatment with two glycosylation inhibitors: tunicamycin and benzyl-alpha-N-acetylgalactosamine (BAG), MUC1/KL-6 expression was significantly reduced in iCC cells, cell adhesive properties were decreased and cell invasive abilities were significantly limited (after BAG treatment) [112]. This indicates that MUC1/KL-6 is responsible for adhesion of iCC cells and their invasive properties.

Studies by Bozkaya et al., using highly motile and invasive, poorly-differentiated mesenchymal-like HCC cell lines, describe overexpression of MUC1 and c-Met (HGFR). These results suggest that, under basal conditions, MUC1 and c-Met interact with each other. Since c-Met is a cell differentiation marker, co-expression of MUC1 and c-Met in these cell lines might indicate the participation of MUC1 in the EMT process during hepatocarcinogenesis. Additionally, it was proven that the activation of HGF/c-Met signalling pathway targets MUC1 to reduce its protein level. This action prevents the down-regulatory effects of MUC1 on HGF/c-Met signalling and increases motility and invasiveness. The research also indicates that MUC1 is a potential regulator of HGF/c-Met mediated β-catenin activation and of Myc expression in HCC cells [38].

Li et al., with the use of stable MUC1 knockdown in SMMC-7721 cells and global gene expression analysis, showed potential participation of this mucin in hepatocarcinogenesis through regulation of various signalling pathways. When it comes to the Wnt/β-catenin signalling pathway, they have proved that a knockdown of MUC1 expression blocked the translocation of β-catenin from the cytoplasm to the nucleus and inhibited SMMC-7721 proliferation through inhibition of the β-catenin signalling. Other Wnt-responsive genes regulated by MUC1 included cyclin D1, c-Myc, as well as other transcription factors, which resulted in inhibition of cell proliferation, induction of the cell cycle arrest in the S-phase, enhanced apoptosis and significant suppression of tumour growth in vivo. Other genes that were down-regulated more than 2-fold in MUC1-knockdown clones (MR1-C6 and MR1-D4) encompassed such signalling pathways as NF-κB, insulin, TGF-β, MAPK and VEGF [39].

Further works explained the co-operation of MUC1 with target genes/molecules of the TGF-β signalling pathway. It was shown that the MUC1 expression not only enhanced TGF-β1 expression (mRNA and protein) but also increased luciferase activity driven by a TGF-β promoter, as well as elevated the activity of JNK and c-Jun [40,44]. In further research, it was proven that MUC1 overexpression suppresses TβRI-mediated pSmad3C signalling (which involves growth inhibition through up-regulation of p21/WAF1). In the same time, it directly activates JNK to stimulate oncogenic pSmad3L signalling (which fosters cell proliferation by up-regulating c-Myc). High correlation between MUC1 and pSmad3L/c-Myc but not pSmad3C/p21 (WAF1), expression was observed also in tissues from HCC patients [41]. It was also shown that promotion of the migration and invasion of HCC cells occurs due to autocrine TGF-β levels, induced by MUC1 via JNK/AP-1 signalling pathway. TGF-β1 expression levels (mRNA, protein), Smad3L (Ser-213) and MMP-9 were significantly elevated in MUC1-overexpressing cells (Bel-7402-MUC1 and Hep3B-MUC1), compared to the control cells [40]. Participation of MUC1 and JNK/TGF-β signalling in progression and tumorigenesis of HCC was confirmed by Wang et al. [44]. All of these works support the stimulating influence of MUC1 on proliferation, migration and invasiveness of HCC cells, acting through TGF-β signalling pathway and TGF-β-related signalling molecules (Smads) or other transcription factors (c-Jun, c-Myc).

The group of PLC risk factors includes prolonged inflammation and hepatolithiasis [94]. Mucin hypersecretion and alteration of the mucin profile, such as an aberrant expression of MUC2 and MUC5AC (which form large polymers), are important constituents of the lithogenesis of hepatolithiasis. Research conducted on mouse biliary epithelial cell cultures showed 4–5 fold increase in MUC2 and MUC5AC transcripts, compared to control, induced by LPS treatment, which was mediated via TNF-α production and activation of PKC in these cells [115]. Similarly, the work of Ishikawa et al. described aberrant expression of CDX2, which resulted in overexpression of MUC2 in mucinous CCs and intraductal papillary neoplasia of the liver, often associated with hepatolithiasis [116]. Another study confirms the observation that MUC2 and MUC5AC overproduction can be associated with hepatolithiasis and related to inflammation. Yang et al. proved that exogenous PGE2 increased MUC2 and MUC5AC mRNA expression in a dosage-dependent manner, independent of IL-1β and TNF-α, via EP4-p38MAPK signalling pathway [65]. There are also papers suggesting that the loss of MUC2 gene expression is a critical requirement for the development of HCC. Using different HCC cell lines (7721, Huh7 and Hep-G2) it was shown that the expression of MUC2 can be activated by 5-Aza-CdR or TSA (epigenetic inhibitors of MUC2 gene). The effect of 5-Aza-CdR and TSA on MUC2 expression differed depending on the type of HCC cells. The demethylation of MUC2 was found in all HCC cell types treated with both epigenetic inhibitors [52].

Recent studies also point at the role of MUC13 in hepatocarcinogenesis, both in in vitro (seven HCC cell lines and immortalized normal liver cell lines, MIHA and LO2) and mouse in vivo model. Strong oncogenic activity of this mucin includes promotion of cell growth, colony formation, cell migration and tumour formation in nude mice. MUC13 was overexpressed in all HCC cell lines except SMMC7721. MUC13 promoted G_1_/S phase transition through activation of Wnt signalling, binding β-catenin and increasing its phosphorylation at Ser552 and Ser675 sites. This action led nuclear translocation of β-catenin and up-regulation of its downstream target genes (e.g., Axin2, c-Myc and CyclinD1) (Figure 1). Knockdown of AKT with shRNA in MUC13-overexpressing cells nullified the elevated phosphorylation of β-catenin by MUC13 [56].

Contrarily, the studies of MUC15 showed that this mucin negatively regulates metastasis and local growth of HCC cells in vitro, as well as in vivo. Stable expression of this mucin in HCC cell lines (SMMC-7721 and HCC-LM3) reduced their proliferation and invasive features, impeding the ability to form metastatic tumours in mice. This type of mucin may exert its anti-metastatic capabilities by binding EGFR and accelerating EGFR internalization, thereby promoting EGFR degradation and inhibiting EGF-induced PI3K-AKT activation. Additionally, MUC15 regulates expression of MMP-2 and -7 (decrease), as well as of TIMP2 (increase), by blocking PI3K-AKT signalling (Figure 1). Epigenetic regulation is most probably responsible for lowering expression of MUC15 in HCC cells (DNA hypermethylation of the MUC15 promoter) [53].

All these results can suggest that several mucins (MUC1/KL-6, MUC2, MUC5AC, MUC13, MUC15) may be potential targets for HCC treatment. Learning the mechanisms underlying mucins upregulation under inflammatory stimulation, hepatolithiasis and so forth could particularly help in early therapeutic decision making in HCC patients [16,38,39,40,41,42,44,53,112].

### 4.3. Gross Mucin as a Pathological Feature of Biliary Papillary Neoplasms

Mucin production by liver tumour cells was long associated with biliary differentiation. During primary classification, three types of mucin-producing tumours were categorized: papillary CCs, biliary cystadenomas and biliary cystadenocarcinomas [117,118]. Pioneering comparative studies on mucin-producing CCs and non-mucin producing CCs proved that the former occur in 13% of affected and are characterised by longer survival time [45]. It was confirmed that hepatobiliary mucinous cystadenoma and cystadenocarcinomas are the precursors of BT tumours which overproduce and/or oversecrete mucins [75,87,119]. Cystadenomas and cystadenocarcinomas were since reclassified as “cystic intraductal papillary neoplasm of the intrahepatic bile duct” (M-IPNB, IPMN), “biliary intraductal tubulopapillary neoplasms” (b-ITPN) and “hepatic (biliary) mucinous cystic neoplasms” (MCNBs, MCN-L). The presence of an ovarian-like stroma (OLS) has been established to define the diagnosis of MCN-L [90,93,95,96,120,121,122,123].

The latest histopathological classification (2018) details two types of IPNBs: type 1 IPNB (classical IPNB), which is histologically similar to intraductal papillary mucinous neoplasms of pancreas and typically develops in the IBDs and type 2 IPNB (so called papillary carcinoma or cholangiocarcinoma), which has a more complex histological architecture with irregular papillary branching or with foci of solid-tubular components, typically involving EBDs. The classical IPNB has more or less similar features to the IPMN. Hence, gross mucin is common (~80%) in type 1 IPNB and relatively rare in papillary CC (~10%) [124]. It is worth noting that the concept of IPNB is still confusing, requiring continuous research based on already described pathological neoplasm properties [88,124]. A need for revision of WHO classification of all ductal adenocarcinomas and subtypes of IPMN is suggested, also taking validated and recommended diagnostic use of MUC1 antibody, which recognizes fully glycosylated MUC1, into consideration [125].

Cholangiocarcinoma (currently type 2 IPNB) is also anatomically classified into perihilar and distal types of iCC [95,96,121,124]. Hilar carcinoma (called Klatskin tumour) and peripheral CC can be distinguished among iCCs, with both previously noted to exhibit mucin production [95,110]. Both subtypes of iCC are thought to arise from topologically different IBDs [107]. The perihilar type emerges from large-sized perihilar intrahepatic segmental and septal bile ducts, lined with mucous-producing tall cylindrical cholangiocytes [126]. In turn, peripheral type originates in peripheral small-sized IBDs (interlobular bile ducts, cholangioles/ductules and the canals of Hering) [127], which are lined with non-mucous-producing cuboidal cholangiocytes [128,129]. The pathological classification of iCCs divides them into mass-forming (MF), periductal-infiltrating and intraductal-growth types [130]. Perihilar/distal CCs are classified into flat- and nodular-infiltrating, as well as papillary types [95]. Biliary intraepithelial neoplasia (BilIN), a flat lesion, precedes periductal-, flat- and nodular-infiltrating CCs, whereas IPNB precedes the intraductal-growth and papillary CCs [95]. The papillary neoplasms in the biliary tree represent ~4–10% of all biliary epithelial neoplasms [45,85,86,87,131]. Large heterogeneity of these neoplasms causes much effort to be focused on their differential diagnostics, including their relation to pancreatic IPMNs [90,121,124].

There are described cases of MCN-L with biliary communications, as well as MCN-L with intermediate- to high-grade intraepithelial dysplasia with a dilation of the left hepatic duct [132,133], including an interesting report on the simultaneous occurrence of these two histologically distinct entities in the liver (M-IPNB and MCN-L) [134].

#### 4.3.1. Various (Immuno)phenotypes of Mucin-Producing Hepatobiliary Tumours

Quantitative evaluation of mucin production by both types of IPNBs is one of the criteria of the modern classification of these cancers [124]. In turn, determination of mucin type and its cellular localization using IHC method helps to evaluate the histological immunophenotypes of IPNBs. WHO 2010 recognizes four different phenotypes of IPNBs (gastric, intestinal, pancreatobiliary and oncocytic) and recommends analysis of selected mucin (MUC1, MUC2, MUC5AC, MUC6) expression, as well as evaluation of architectural and cell differentiation patterns, for correct classification [88,93,96]. Some sources detail only two immunophenotypes of IPNBs: the “pancreatic” which is more similar to IPMN and the “non-pancreatic” with frequent high-grade dysplasia (papillary CC) [123,135]. Using the hierarchical clustering and differential IHC analysis for adenocarcinomas of the pancreatobiliary system, three IHC tumour subtypes were identified, namely: extrahepatic pancreatobiliary, intestinal and intrahepatic CC [136].

In 2018 classification, type 1 IPNB exhibits gastric or oncocytic phenotype, although pancreatobiliary and intestinal phenotypes are also seen. Type 2 IPNB (papillary CC) is typically of pancreatobiliary or intestinal type. As mentioned, high amount of mucin is only common in type 1 IPNB. However, lack of mucin overproduction does not exclude the possibility of the classification into this type [124].

In further literature review, histological classifications that were valid during the time of source publication were left unaltered. Hence, for example the IPMN term used in the cited works describes what is now known as “classical” IPNB (type 1) [124].

Numerous studies describe a detailed profile/pattern of mucins in PLC, which eases their classification, differential characteristics and prognostics [88,90,119,123,137,138,139]. While MUC1 expression dominates in the pancreatobiliary phenotype, its small amounts are also detected in gastric, as well as other (intestinal, oncocytic), phenotypes of IPNB [87,88,140]. It is thought that aberrant expression of MUC1 can lead to IPNB invasion, leading to tubular adenocarcinoma [87].

MUC2 is a mucin characteristic for intestinal phenotype of IPNBs. Its aberrant expression may lead to development of intestinal metaplasia and the invasion of IPNB, leading to mucinous carcinoma [87,88,140].

MUC6 expression is observed in gastric and oncocytic phenotypes, while MUC5AC can be observed in either of four phenotypes [88]. The expression pattern described as “MUC1+/MUC2-/MUC5AC+” was characteristic for CCs and pancreatic ductal adenocarcinomas and could distinguish CC from HCC (negative for all mucins) [137]. In turn, comparison of differentially located liver cystadenomas (16 arising in the liver and 4 in the EBDs), showed pancreatobiliary (CK7+, CK19+, MUC1+), as well as intestinal differentiation (CDX2+, CD20+, MUC2+) with OLS features [119].

Immunophenotype characterization of different mucinous BTCs, showed that the gastro-intestinal phenotype (MUC2+, MUC5AC+ and MUC6+) was more frequently seen in MCN-L and IPMN than in benign mucinous cystadenomas. Additionally, the expression of MUC1, MUC2, MUC5AC and MUC6 was significantly more common in IPMN than in MCN-L. Differential expression of MUC1 was observed only in malignant cases of both types of tumours [139]. Other authors confirmed higher mucin production in IPNBs (72%) (currently type 1 IPNB) compared to the papillary CCs (7%) (type 2 IPNB). Gastric-type and oncocytic-type tumours were only detected in IPNBs (type 1). The expression of MUC1, MUC2 and MUC6 differed significantly between both types of IPNBs and non-papillary CCs [123]. Results concerning gross mucin expression support one of the main criteria of IPNB classification [124], while studies of particular apomucins allowed for more detailed differential diagnostics of both subtypes. Another study comparing different CCs showed that, among ICCs, ~61% only contained mucin-producing CC features (M-CCs). Others displayed histological diversity, showing focal hepatocytic differentiation, as well as ductular areas (mixed-CCs). Expression of MUC1 was significantly up-regulated in hilar CCs and M-CCs, compared with mixed-CCs and CoCCs. Study on biliary/HPCs and hepatocytic markers could indicate cellular origin of iCC subtypes (Muc-CCs and hilar CCs form mucin-producing cholangiocytes, whereas mixed-CCs and CoCCs from HPCs) [107].

In the Far East, in aetiology of iCC and HCC, a large role is played by HBV and HCV infections [99,105,128,141,142]. It was shown that N-cadherin was a marker of iCC subgroup associated with viral hepatitis, characterized by cholangiolar differentiation. MUC2 expression was more frequently found in N-cadherin-negative CCs, while the expression of MUC1 was similar in both groups of tumours [128].

Rates of mucus secretion and ductal MUC1 expression were also compared between iCC and HCC, showing higher rate and expression of MUC1 in the former [143].

Mucin expression was also analysed in different groups of biliary IPN and pancreatic IPMN. Mucin hypersecretion was significantly less frequent in type 1 and 2 IPNB (50%, 15.3%, respectively) than in IPMN of the pancreas (83%). It was shown that the intestinal subtype of IPNB was more frequently positive for MUC1 and less frequently positive for MUC2, MUC5AC and CDX2 (a MUC2 regulating transcription factor), compared to each subtype of pancreatic IPMN [90].

#### 4.3.2. Differential Tissue Expression of Mucins in Primary Liver Cancers

The dominating mucin type, expressed in most PLC cases, is undoubtedly MUC1 (pan-epithelial mucin), with other detected mucins including MUC2, MUC3, MUC5AC and MUC6. Expression of these mucins is detected in pre-cancerous lesions of liver and BT, CCs, cHCC-CC, CoCC, as well as in HCC [81,82,84,86,87,88,96,112,123,135,138,144]. The subcellular localization of MUC1 in PLC includes both cell membranes and the cytoplasm of neoplastic liver and/or biliary cells [32,110] (Figure 2).

Among precursors and early lesions of CCs, MUC1 positive staining was observed in Meyenburg complex (VMC)-like duct, with MUC6 expression detected in Epithelial Membranous Antigen (EMA)-luminal type of bile duct adenoma (BDA). Both of these lesions arose in chronic liver diseases [145]. In chronic B and/or C hepatitis, increased expression of MUC6 was observed in proliferating bile ductules and intralobular S cells, which correlated with the degree of active necroinflammation [74].

Analysis of different glycoforms of MUC1 in various cystic liver diseases allowed for suggestions concerning cystogenesis in the liver, from IBDs through biliary microhamartomas, to hepatic cysts. Common expression of „normal” MUC1 in epithelial cells lining cysts and IBDs, suggests that cholangiocytes expressing MUC1 are related to the whole cystogenic process in the studied diseases. The highly-glycosylated glycoform of MUC1 (“mature” form) present in ~half of hepatic cysts studied, may be related to the late cystogenetic process in liver cystogenesis [81]. Later research on biliary cystic tumours with bile duct communication, show expression of MUC1 and MUC2 in the neoplastic biliary epithelium in most of the cases [86]. Other literature data describe strong mucin expression (alcian blue-positive) in all of the cases of cystic-micropapillary (C-P) lesions of PBGs and only some of the cystic lesions. MUC1 expression was negative in C-P and cystic lesions, while that of the MUC5AC was more common in C-P than in cystic lesions of PBGs. This suggest that the C-P lesions of PBGs may be the precursors of biliary epithelial neoplasms [89]. Yonezawa et al. had reported that increase in MUC1 expression was correlated with increasing grades of biliary intraepithelial neoplasia (BilIN). The high expression of MUC1 was observed in iCCs. Negative expression of MUC2 in any grades of BilINs, could help in differentiating BilIN from mucin-producing bile duct tumour (MPBT)-columnar type/IPNB-intestinal type. High expression of MUC4 considered iCCs and carcinomas of EBDs. MUC5AC expression was present in all of the studied pancreatic and biliary neoplastic tissues [146].

Studies of Sasaki et al. concerning CCs showed strong expression of MUC1, with only focal MUC2 expression. Common and aberrant expression of MUC5/6 in biliary epithelial dysplasia and in CCs, suggests that biliary epithelial cells gain a gastric phenotype during carcinogenesis [84]. In studies of Higashi et al., expression of MUC1 was more common in mass forming (MF) type of CCs (85–95%) than in bile duct cyst adenocarcinoma (BDCC). In contrast, BDCC, intraductal growth type and/or periductal infiltrating type of CCs showed significantly higher MUC2 expression rates than MF-CCs (86%, 67%, 25%, respectively) [47]. Other research, comparing different subtypes of CCs, confirmed that the most abundantly represented mucins in these cancers were MUC1 (~66%) and MUC5AC (~61%), followed by MUC2 (~24%) and MUC6 (~14%) [144]. Mall et al. showed extensive expression of MUC1 (>50%) in 44% and MUC5AC in 46% of CCs cases. Extensive MUC1 expression was correlated with CC metastasis [138]. Descriptions of some of the clinical cases indicate coexistence of strong MUC5AC-positivity in perihilar CCs and extensive MUC6-positivity in C-P neoplasm of PBGs, suggesting different origin of these neoplasms [50].

Mucins can also be produced by cHCC-CCs - tumours characterized by large invasiveness and short overall survival (OS) [46,97,147]. In some of the clinical cHCC-CC cases, mucin production was observed in both the HCC and the CC areas of these tumours [147]. Ng et al. described mucin production in 76% of cHCC-CC [46], while in research of Xu et al., KL-6/MUC1 expression was present in all CCs and CC areas of cHCC-CC tissues but not in HCC area [112]. Similarly, Xu et al. showed glandular formation accompanied by mucin production, representing iCC areas in cHCC-CC, in more than 80% of the cases [51]. A clinical case of HBV-positive patient with cHCC-CC was described, with positive expression of MUC1, MUC2, MUC5AC and MUC6, as well as stem cells features and DP malformations [99]. Apical membrane colocalization of MECA-79 sulphated glycans and MUC1 was reported in cholangiolocellular carcinoma (CoCC), which indicates that MUC1 serves as a scaffold protein for MECA-79 sulphated glycans. This would confirm the cholangiolar/ductular differentiation, with the possibility that MUC1 could serve as a useful marker of this subtype of cHCC-CC [110].

Mucin production was also reported in hepatolithiasis [75,80,84,85,86,116]. Focal expression of MUC1 (33% cases), MUC2 (64%) and MUC5 (89%) was detected in large IBDs with hepatolithiasis. Strong expression of MUC3 and MUC6, as well as focal expression of MUC2 and MUC5, were accompanied by markedly proliferated intramural and extramural PBGs. According to authors, more commonly represented mucins, for example MUC2 and MUC5, could be involved in the pathogenesis of hepatolithiasis [75]. Other authors characterized three different mucin patterns in cholangiocarcinogenetic pathways in hepatolithiasis. Increased expression of MUC1 in BilIN (MUC2−/CK7+/CK20− pattern) and IPNB (MUC2+/CK7+/CK20+) is associated with tubular adenocarcinoma, while colloid carcinoma in IPNB is characterized by MUC1-negativity and less advanced pathologic stages [86].

### 4.4. Serum/Bile Levels of Mucins in Primary Liver Cancers

With introduction of molecular techniques, it is now possible to examine the role of mucins as diagnostic-prognostic markers not only in tissues but also in serum and/or bile [33]. Such studies in PLC concern mostly MUC1 and MUC5AC. Human MUC1/KL-6 in HCC patients was even described as a tumour marker. Significantly higher mean serum levels of KL-6 in HCC (556 ± 467 U/L) were detected, in comparison with non-HCC groups either with (391 ± 176 U/L) or without (361 ± 161 U/L) liver cirrhosis [148]. Significantly elevated KL-6 serum level was also observed in older (˃60 years) HCV-positive patients with HCC. The HCC rate was higher (37.5%) in the patients with elevated KL-6 than with normal KL-6 group (~8%) [149]. When the MUC1/KL-6 levels in sera from various liver cancer patients were compared, significantly higher levels were noted in CC patients than in HCC, metastatic and healthy individuals. The cut-of value of 248 U/mL could distinguish CC patients from HCC patients [150]. Other multicentre studies in Japan indicate *Wisteria floribunda* agglutinin (WFA)-sialylated MUC1 as a new glycoprotein marker of CC. Higher levels of WFA-MUC1 were observed in BTC/iCC than in benign BT diseases. Superiority of this form of MUC1 over carbohydrate antigen 19-9 (CA19-9) and CEA was proven, when used for the purpose of differential diagnostics of benign and malignant diseases of the biliary pathways, as well as for stage I and II carcinomas [151].

With the use of immunoblotting marked with antibody against MUC5AC, Wongkham et al. detected this type of mucin in ~62% of CC patients, as compared with only 3% of patients with benign hepatobiliary diseases, 10% with hepato-gastrointestinal cancer and none of the healthy control. Additionally, serum MUC5AC correlated with the tissue expression in CC patients [152]. Use of the ELISA technique for detection of MUC5AC, allowed for achieving 71% sensitivity and 90% specificity for discriminating CC patients from the controls [153]. In other study, MUC5AC serum expression was found more frequently in BTC (44% of patients) than in PSC (14%) [49]. Additionally, serum MUC5AC was associated with patients with blood type A, larger-sized tumours (>5 cm) and advanced-stage disease [154]. Quantitative research showed significantly higher serum MUC5AC levels in BTC patients, compared to those affected with benign biliary disease [155,156]. Additionally, serum MUC5AC ≥ 14 ng/mL was associated with lymph-node metastasis, tumour stage (IVb) and a worse prognosis in BTC patients who underwent surgery, compared with patients with lower levels of this mucin [155]. Immunoblot analysis confirmed the presence of stronger MUC5AC expression in serum than in bile of CC patients [156]. Serum MUC5AC levels gave 60.6% sensitivity and 82.3% specificity at a cut-off of >0.67 ng/mL, while a panel combining PKM2, CYFRA21.1, MUC5AC and GGT was beneficial in differentiating malignant (CC) from benign biliary disease (PSC), warranting validation in a prospective trial [157]. From the tested markers, only the serum levels of MUC5AC were significantly correlated with BTC presence [155].

Based on over 1200 patients, the meta-analysis conducted by Xuan et al. confirmed that serum MUC5AC is a good diagnostic marker in CCs. However, targeting MUC5AC epitope has a higher pooled sensitivity than targeting MUC5AC protein (0.77 vs 0.63) [158]. Effectiveness of serum MUC5AC levels (OR = 4.52) and immunoblot levels (OR = 2.61) as diagnostic biomarkers in Thai population (OR = 8.32) was confirmed in another meta-analysis [159].

Few works consider expression of mucins (e.g., MUC4, MUC5AC, MUC1/KL-6) in bile of BTC patients, indicating their role in diagnosis and therapeutic strategies. Matull et al. showed a 1.9-fold increase (95% CI: 1.69–2.33) in MUC4 mRNA and a 3.8-fold increase (95% CI: 3.33–4.43) in MUC5AC mRNA expression levels in BTC bile samples, compared to benign biliary conditions (95% CI: 0.86–1.16; 95% CI: 0.83–1.21, respectively) [49]. Additionally, evaluation of serum to bile ratio of MUC5AC (better than serum level alone) showed excellent diagnostic performance for differentiating CC from cholangitis and cholelithiasis [156]. Significantly higher concentrations of MUC2 and MUC5AC was also detected in bile samples from the hepatic ducts affected by intrahepatic calculi, as compared with the unaffected hepatic duct of patients with hepatolithiasis [65]. Recent study of Onoyama et al. showed increased KL-6 concentration of bile (34.6 ± 51.6 U/mL) in BTC, compared to benign biliary disease (5.2 ± 3.9 U/mL). The authors noted that bile KL-6 concentration to bile cytology measurements, the sensitivity for the diagnosis of BTC was increased significantly [160].

### 4.5. Mucins as Prognostic Factors in Primary Liver Cancers

While most of the clinical results indicates prognostic role of MUC1 and MUC2 expression [161], the number of works pointing at a role of other mucins (e.g., MUC4, MUC5AC, MUC13, MUC15, MUC16) in PLC increases [53,54,64,162,163].

Singular studies conducted on different subtypes of CC (invasive iCC, MF-CC; extrahepatic CC) [47,76,82,145,164], as well as extensive meta-analysis (3425 patients) confirmed that increased MUC1 expression (including that of its more sialylated glycoform) was significantly associated with poor OS in patients with GI tract carcinomas, including CC [in fixed-effect model (FEM): HR = 2.52, 95%CI: 1.42–4.49, *P*_FEM_ = 0.252; and at random-effect model (REM): HR = 2.34, 95%CI: 1.30–4.22, *P*_REM_ = 0.244] [161]. Additionally, it was proven that patients with the cytoplasmic pattern of MUC1 expression showed significantly lower survival rates [48,82]. MUC1 overproduction was most commonly detected in poorly differentiated CCs, correlating with T category, gross type of intra- and extrahepatic CC [144], metastasis of lymph nodes, portal canal emboli and post-operational recurrence of the carcinomas. MUC1-positive patients showed faster metastasis, as well as shorter OS, compared to those that were MUC-1 negative [82].

As opposed to MUC1, positive expression of non-sialylated forms of MUC2 in two subtypes of iCC was a favourable prognostic indicator [47]. In turn, Ling et al. indicated that decreased expression of MUC2 (−ΔΔCt < 0) was significantly correlated with poor OS (HR = 0.238, 95% CI: 0.13–0.43). Moreover, MUC2 mRNA and promoter methylation significantly correlated with OS after surgery in HCC patients [52].

When it comes to other mucin types, it was demonstrated that tissue expression of MUC4 can be a significant independent factor for poor prognosis and is a useful marker to predict the outcome of the patients with iCC-MF [76] and EBD carcinoma [162,163,164]. The survival of the patients with high MUC4 expression was significantly worse than that of the patients with low expression. Only in iCC-MF, double positive expression of MUC4 and ErbB2 showed a shorter cumulative survival rate compared to non-expressing tumours [76]. Large meta-analysis (~2000 patients) concerning serum MUC4 levels confirmed usefulness of this prognostic marker in cancers, including BTC. Significant association was found between elevated MUC4 expression and poorer OS (HR: 2.41, 95% CI: 1.69–3.42) [162].

It was also proven that serum MUC5AC could be a marker of bad prognosis in CC patients [154,155]. Patients with positive serum MUC5AC had a 2.5-fold higher risk of death compared to those that were deemed MUC5AC-negative. Additionally, positive serum MUC5AC was associated with more advanced-stage of the disease and larger tumours (>5 cm) [154]. Quantitative research showed significantly higher serum MUC5AC levels in BTC patients, compared with benign biliary disease [155,156]. Additionally, serum MUC5AC ≥ 14 ng/mL was associated with lymph-node metastasis, tumour stage (IVb) and a worse prognosis in BTC patients who underwent surgery, compared to patients with lower levels of this mucin [155]. The results of the abovementioned research are confirmed by a meta-analysis, proving that MUC5AC is a promising prognostic factor for cancer, especially for that of biliary and GI origin and is more suitable for predicting cancer prognoses in Asians. The overexpression of MUC5AC was found to be significantly associated with a poor prognosis in the biliary subgroup (HR: 1.83, 95%CI: 1.269–2.639) and the GI group (HR: 1.44, 95%CI: 1.069–1.949) [163].

In addition, MUC13 and MUC15 seem to be new, promising prognostic factors in HCC [53,54,56]. In the case of MUC13, overexpression of this type mucin was significantly associated with poor OS and poor DFS rate [56]. When it comes to MUC15, lower tissue expression was significantly correlated with shorter OS and relapse time [53], as well as poor clinical prognosis of the patients [54].

MUC16 expression was positive in 48% of the MF-CC samples and was indicated as a prognostic factor of poor survival in patients [64].

### 4.6. Mucins in Liver Cirrhosis

The role of mucins in liver cirrhosis is mostly confirmed by clinical works, usually concerning MUC1 [80,82,165]. In studies of mucin expression in various hepatobiliary diseases, MUC1 was expressed in different frequency in all types of pathological lesions, especially in cases of destructive cholangitis in PBC and hepatic duct injuries in chronic hepatitis [80]. Yuan et al. detected MUC1 expression in only 4/20 cirrhotic liver tissues, with lower expression observed in cirrhotic tissue, compared to that affected with PLC [82]. In other study, the levels of both MUC1 and c-Met expression in cirrhotic samples were significantly higher than in normal tissue and lower than in HCC samples. Cirrhotic liver tissue showed weak, diffuse cytoplasmic MUC1 staining [38].

Elevated serum MUC1/KL-6 expression was observed in patients with various chronic liver diseases, mainly associated with HCV infection [166]. Higher levels of KL-6 were observed in cirrhotic patients (377.6 ± 212.1 U/mL) than in chronic hepatitis patients (283.5 ± 131.4 U/mL), both without interstitial pneumonia. The authors concluded that KL-6 reflects hepatic fibrosis better than pulmonary fibrosis [166].

Elevated MUC16 (CA125) levels were also reported in patients with liver cirrhosis associated with ascites [165,167]. A retrospective study of Edula et al. noted CA125 levels elevation in 85% of adult patients with cirrhosis. The higher level of CA125 was correlated with more advanced degree of liver decompensation [165,168].

### 4.7. Mucins in Hepatocellular Carcinoma – In Vivo Studies

In the current classification of HCC, based on molecular features, three categories of HCC are distinguished, including proliferation-progenitor, proliferation-TGF-β and Wnt-catenin β1. The proliferation-progenitor group of HCC is generally associated with more aggressive phenotype and poor outcome of patients [78]. This classification does not take quantitatively or qualitatively altered mucin production into account. Based on histological HCC properties, three main forms of this cancer are recognized: pure, mixed and motley [168]. Evaluation of mucin expression turned out to be very useful in morphological classification of the subtypes of this heterogenous cancer in human, especially when it comes to mixed/combined HCC-CC. In WHO 2010 classification, cHCC-CCs are divided into classic and those with stem cell features. The classic cHCC-CC has typical HCC and CC areas [98]. For the CC area, its IHC profile includes positive staining for mucin, which is essential to demonstrate the biliary component, CK7, CK19 and AE1 [98,169]. According to the recently published research on genetic characteristics of early stage HBV-associated HCC, among the eight most commonly mutated genes in this cancer is *MUC16*. The mutation was heterozygous, leading to sequence changes of the corresponding aa in the encoded protein [170].

Considering tissue expression of different mucins in HCC, MUC1, MUC2 and MUC5AC were most commonly investigated [38,52,82,83,137]. Currently, other mucins are also included in studies, for example MUC13 or MUC15 [42,54,56]. Studies of Lau et al., conducted on different types of primary cancers, did not show MUC1, MUC2 or MUC5AC positive expression in HCC [137]. Later works described increased expression of the MUC2 gene in only ~31% of the HCC patients, with mean expression lower (mean _−ΔCt_ = −4.70; 95% CI, −5.88–−3.53) than that in non-HCC tissues (mean _−ΔCt_ = −2.98; 95% CI, −3.99–−1.97). Moreover, the MUC2 mRNA expression was lower in HCC patients with MUC2 promoter hypermethylation. It was also reported that the MUC2 promoter was significantly more often hypermethylated in HCCs, than in non-tumour samples (~62% vs. ~19%, respectively). From a clinical point of view, the negative correlations between MUC2 mRNA, HBV viral load and AFP in HCC were especially important. The study suggests that the loss of MUC2 mRNA and hypermethylation could be poor prognostic factors in HCC [52].

Comparisons of mucin expression in HCC and CC gave mixed results [38,82,83]. In one of the studies, no significant differences between these two cancer types were found, suggesting that MUC1 gene expression was not associated with histological classification of the hepatic tumours [82]. Meanwhile, analysis of KL-6/MUC1 expression in patients with CC, HCC and cHCC-CC, allowed to distinguish CC (positive in all cases) and cHCC-CC (positive in CC areas) from HCC (negative in all cases) [83]. In the study of Bozkaya et al. the expression levels of MUC1 and c-Met in HCC were greater than in both normal and cirrhotic liver tissues. MUC1 staining was mainly cytoplasmic in tumour cells. [38]. Other authors reported presence of the extracellular myxoid/mucinous material in hepatic adenomas and HCCs. Extraordinary morphology of these mucin producing hepatic tumours, their characteristic clinical properties, as well as lack of morphological and IHC properties of biliary differentiation, suggest presence of unique variants of these PLCs [171].

In the current literature, we can find confirmation of these suggestions, as mucin producing was also described in HCC that contained intracellular and extracellular myxoid matrix without evidence of biliary differentiation. This study confirmed the expression of MUC5AC and MUC6, as well as negative expression of MUC1 and MUC2 throughout the tumour, with the final diagnosis indicating mucin producing HCC [55]. Contrary to Salaria et al., [171] who only detected myxoid material extracellularly, this research mostly localized mucins intracellularly (in a contained region of the tumour), confirming the possibility of production of these glycoproteins by the HCC cells. Authors discuss the possibility of detection of mutated mucins produced by HCC cells simultaneously exhibiting biliary cells markers (CK7 and CK19) [55].

Finally, there are studies on classification of mucin-producing PLC (including HCC), which aim to improve differential diagnostics, better characterize cellular origin of these heterogenous tumours, as well as ease therapeutic decisions. It is known that cHCC-CC exhibits poorer prognosis than HCC, as well as different treatment modalities [168]. Evaluation of MUC1 expression is used as one of the cholangiocyte markers in the CC component. It was reported that ordinary HCC includes cholangiocyte marker-positive areas [172]. Research on HCC lesions no larger than 5 cm showed presence of cholangiocyte markers: CK7 (in 75%), CK19 (22%), the hepatic stem/progenitor cell marker (C-kit – ~12%) and mucin production (MUC1 - ~12%). The small-size of cells positive for these markers and formation of small foci, could rather suggest transdifferentiation of HCC cells than malignant transformation of stem/progenitor cells [172]. Later studies of the same group confirmed the expression of MUC1 in HCC. Depending on the size of the studied HCCs, a significant difference in positive ratio of MUC1 was observed between S3 HCC group (tumour size 5.0–8.0 cm) (~46%) and both S1 (tumour size <2.0 cm) (~7.0%) and S2 groups (2.0–5.0 cm) (~9.0%). The positive ratio of MUC1 of the poorly differentiated HCC group was significantly higher than those of the well-differentiated and moderately differentiated HCC groups. The authors concluded that HCC can acquire the markers positive for cholangiocytes and stem/progenitor cells, during HCC progression [173]. Results of both works support the transdifferentiation process as a formation mechanism of the classical type of cHCC-CC [172,173]. Other authors describe dual tumour phenotype in patients with classical HCC that expresses cholangiocyte markers. This subtype of the tumour, present in ~10% of total HCCs, exhibits highly malignant behaviour (lower OS and RFS), compared to pure HCC. The combination of IHC intensities of CK19 and MUC1 was significantly associated with tumour size, microvascular invasion and satellite nodule formation [174].

There are two other mucins in the group of new, promising prognostic HCC markers: MUC13 [56] and MUC15 [53,54]. In the case of MUC13, overexpression was detected in 44% of HCC cases and was significantly associated with tumour size, stage, encapsulation, venous invasion and poor outcome. The overexpression of MUC13 was significantly associated with HBV DNA copy number. Moreover, MUC13 overexpression was significantly associated with nuclear translocation of β-catenin in HCC samples, playing a pivotal role in the development and progression of HCC through activation of Wnt signalling pathway [56]. When it comes to MUC15, statistically significant decrease in expression of this mucin was noted in most of the HCC patients, compared to adjacent non-tumour tissue [53,54]. In addition, lowered expression of MUC15 was significantly correlated with the TNM stage, intrahepatic or lymphatic metastasis, portal vein thrombosis [54], high levels of AFP, vascular invasion, lack of encapsulation [53] and poor tumour differentiation [53,54]. Decreased MUC15 expression in HCC was a factor of poor clinical prognosis in HCC patients [53,54].

Summary of the results on tissue expression, serum/bile level of mucins in precursors and early lesions of CC, HCC, as well as other types of PLC, with their possible role in pathogenesis, diagnosis and prognosis, were presented in Table 1.

## 5. The Main Headlines of the Review and Conclusions

The research of mucin expression in precursor lesions of CC, HCC and other subtypes of PLC, conducted on in vivo and in vitro models allowed for:indication that although the main producer of mucins in PLC are biliary epithelial cells (cholangiocytes), these glycoproteins can also be produced by neoplastic hepatocytes;proving that, during hepatobiliary carcinogenesis, some mucins are aberrantly overexpressed (e.g., MUC1), while expression of other mucins decreases (e.g., MUC2);characterization of changes in mucins subcellular localization (e.g., cytoplasmic expression of MUC1 in place of membranous) accompanying carcinogenesis;demonstration of the qualitative glycoprotein changes, including different patterns of mucin glycosylation in hepatobiliary carcinogenesis (e.g., sialylation of MUC1 associated with invasive growth of neoplasm);defining that the malignant transformation of liver cells (cholangiocytes, hepatocytes) is associated with oncogenic functions of human mucins (e.g., MUC1, MUC13, MUC15);closer examination of the mechanisms of mucin action, as well as their participation in the most important pathways of hepatic carcinogenesis;characterization of the stages of liver cystogenesis;determination of the role of mucins in the initiation and progression of PLC;immunophenotype characterization of different types of hepatobiliary mucinous lesions, starting from precursor lesions, ending on invasive PLC (e.g., gain of gastric apomucin phenotype in iCC during carcinogenesis);hypothesizing on the cellular origins of rare PLC types;gaining the possibility for early PLC detection in a high-risk groups or suspected patients;determination of histo- and immunohistochemical panel of mucins of the biggest diagnostic-prognostic role in PLC (including HCC);improvement of modern PLC classification, including detailing of the hepatic tumour subtypes.

Concluding, it can be stated that the evaluation of mucin expression in primary liver cancer has both research and clinical value. Mucins may act as oncogenes and tumour-promoting (e.g., MUC1), and/or tumour-suppressing factors (e.g., MUC15). Given their role in promoting PLC progression, both classic (MUC1, MUC2, MUC4, MUC5AC, MUC6) and currently tested mucins (e.g., MUC13, MUC15, MUC16) have been proposed to be important diagnostic and prognostic markers. Estimation of mucin expression and explaining the mechanisms of action requires further studies also in the context of new targeted therapies.

## Figures and Tables

**Figure 1 ijms-20-01288-f001:**
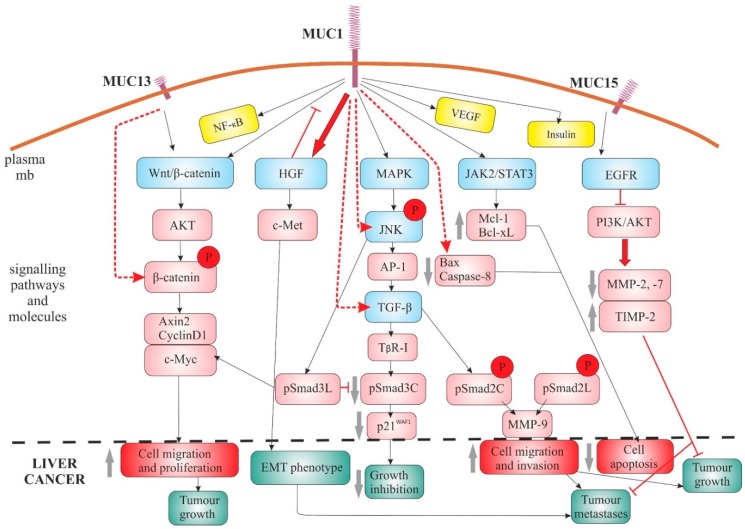
Proposed model of two trans-membrane mucins (MUC1, MUC13) action in promoting hepatocellular carcinoma (HCC) growth, tumour progression and metastasis, as well as the role of MUC15 in inhibition of tumour growth and metastases (with own modifications) [38,39,40,41,42,53,56]. The signalling pathways downstream of the activated Wnt/β-catenin, MAPK with JNK/AP-1, TGF-β with JNK/pSmad3L/c-Myc and JAK2/STAT3 are known to stimulate cancer cell proliferation, survival, migration, invasion and inhibit cell apoptosis; whereas TβRI/pSmad3C/p21^WAF1^ is tumour suppressive signalling. The inhibitory effect of MUC15 on PI3K/AKT signalling pathway is linked to negative regulation of metastasis and local growth of HCC cells. Legend: ⇓—regulation; ⇣—direct binding; ↑/↓—increase/decrease; Ʇ—inhibition.

**Figure 2 ijms-20-01288-f002:**
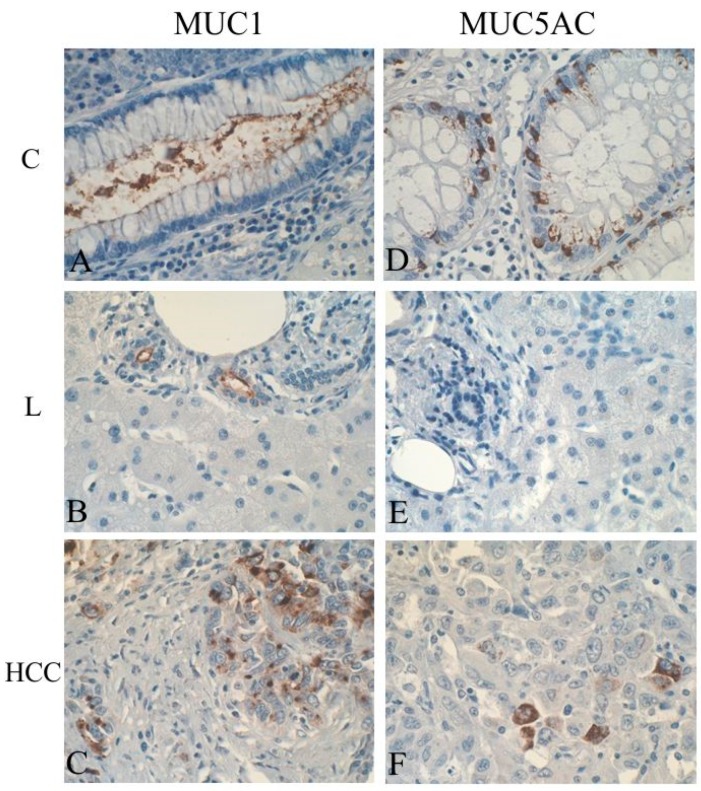
Immunohistochemical localization of MUC1 and MUC5AC in the control colon (C), control liver (L) and hepatocellular carcinoma (HCC) samples. (**A**) MUC1 membranous localization in goblet cells of the normal colonic crypt; (**B**) MUC1-immunopositive biliary cells of interlobular ducts of the normal liver; (**C**) cytoplasmic expression of MUC1 in HCC neoplastic cells; (**D**) cytoplasmic localization of MUC5AC in numerous goblet cells of the control colonic crypts; (**E**) negative IHC reaction for MUC5AC in normal liver; (**F**) MUC5AC-immunopositive scattered neoplastic HCC cells. New polymer-based immunohistochemistry with 3,3′-Diaminobenzidine (DAB) as a chromogen. Haematoxylin counterstained. Objective × 40 (**A**–**F**) (our unpublished data).

**Table 1 ijms-20-01288-t001:** Summary of results regarding tissue (T) expression, serum (S) and bile (B) concentration of mucins in precursors and early lesions of CC, biliary cystic tumours, CCs, HCCs, as well as other types of PLC in vivo with their possible role in pathogenesis, diagnosis and prognosis.

Type of Mucin (A), Source of Mucin (B), Score of Expression (C)			Type of Hepatobiliary Tract Pathology [No of Ref.]						
A	B	C	Precursors of CC *	Biliary cystic tumours **	Cholangiocarcinomas	Hepato- lithiasis	Liver cirrhosis	HCCs	cHCC-CCs, coCCs
**MUC1**	**T**	**+1**	[90,123,146]	[86,145,146]	[32,83,123]	[80]	[82]	[38,172], ♦ [173]	[32,83,99,109,110,112], ↑# [174]
		**+2**	[75]	[47,131]	[88,138]	[75]	[38]	[38,173]	nd
		**+3**	[80]	[81]	[47,146], ↑# [48], ↑♦,# [82], C# [48,82], [84,114], ↑♣,♦ [144], ↑# [161,164]	nd	[80]	[38], ↑♦,# [82]	[84]
		**(−)**	nd		nd	nd		[32,55,83,112,137]	nd
	**S**	**(+)**	nt		↑ [150,151]	nt	↑ [165]	↑ [148,167]	nt
	**B**	**(+)**			↑ [160]		nt		
**MUC2**	**T**	**+1**	[90,116,123]	[86]	[84,88,123,138,144]	[116]	nd		[99]
		**+2**	[85]	[131]	[47]	nd		[52]	[84]
		**+3**	[75]	[47,146]	[47,146]	[75]	nd		
		**(−)**	[86,146]	nd	↓# [47,52]	nd		[55,137], ↓# [52]	nd
	**S/B**	**(+)**	nt						
**MUC3**	**T**	**+1**	[75,84,138]	nd	[84]	[75]	nd		
		**+2**	nd						[84]
		**+3**	nd		[84]	[84]	nd		
		**(−)**	nd						
	**S/B**	**(+)**	nt						
**MUC4**	**T**	**+1**	[146]	nd	[76,138]	nd			
		**+2**	nd						
		**+3**	nd		[146], ↑# [76,164]	nd			
		**(−)**	nd						
	**S**	**(+)**	nt		↑# [162]	nt			
	**B**	**(+)**			↑ [49]				
**MUC5AC**	**T**	**+1**	[90,116]	[131,139]	[84,88,123,128]	nd		[55]	nd
		**+2**	[85]	nd	[138]	nd			
		**+3**	[75,123,146]	[89,146]	[84,144,146]	nd			
		**(−)**	nd		nd	nd		[137]	nd
	**S**	**(+)**	nt		↑ [49,152,154,155,156,158,159], ↑♦ [154,155], ↑# [155,163]	nt			
	**B**	**(+)**			↑ [49]				
**MUC6**	**T**	**+1**	[74,75]	[131,139]	[88,123,138,144]	[75]	nd	[55]	[84,99]
		**+2**	nd		nd	[85]	nd		
		**+3**	[123]	nd	[84]		nd		
		**(−)**	nd						
	**S/B**	**+**	nt						
**MUC13**	**T**	**+1**	nd						
		**+2**	nd					[56]	nd
		**+3**	nd					↑♣,# [56]	nd
		**(−)**	nd						
	**S/B**	**(+)**	nt						
**MUC15**	**T**	**+1;+2;+3**	nd						
		**(−)**	nd					↓♣,♦,# [53,54]	nd
	**S/B**	**(+)**	nt						
**MUC16**	**T**	**+1**	nd						
		**+2**	nd		[64]	nd			
		**+3**	nd		↑# [64]	nd			
		**(−)**	nd						
	**S**	**(+)**	nt				↑ [165,167], ↑♦ [165]	nt	
	**B**	**(+)**	nt						

**Legend**: * including chronic hepatitis, BilIN, IPNB, ITPN-B, MCNB, BDA, biliary adenofibroma and biliary microhamartoma; ** including cystadenomas and cystadenocarcinomas; (+) positive S/B concentration; +1<25% cases with positive T expression; +2≥25–50% positive cases; +3>50% positive cases; (−)—negative expression; C—cytoplasmic expression; ↑/↓ – significant increased/decreased T/S concentration as related to control groups; ♣—significant association between mucin expression and cancer differentiation; ♦—association between mucin expression and more advanced clinical stage of cancer (TNM, tumour size, venous infiltration, microsatellite nodules, metastases, etc.); #—significant correlation with poor prognosis (OS, DFS) and/or cancer recurrence; CC—cholangiocarcinoma; cHCC-CC—combined HCC-CC; coCC—cholangiolocellular carcinoma; HCC—hepatocellular carcinoma; nd—no data; nt—no tested; no of ref.—number of references in order to citation (for details, see text).

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
