# Peer review of "Mucins: the Old, the New and the Promising Factors in Hepatobiliary Carcinogenesis"

_ijms, 2019, doi:10.3390/ijms20061288_

Round 1

Reviewer 1 Report

The manuscript named “Mucins – the old, the new, and the promising factors in hepatobiliary carcinogenesis” reviews studies dealing with the expression and the role of various mucins in normal liver and hepatobiliary carcinogenesis. It is a really deep review, well-written and up-to-date. Figure 1 illustrates very clearly the “dual” effect of various mucins in cancer progression. Figure 2 presents IHC localization of mucins in colon and liver samples. However, the manuscript has little shortcomings. The authors use many font types throughout the manuscript. There are few abbreviations used without previous explanation (f. e. SEA, EMT, EGF, VEGF, NF-kappaB, HGF/cMet, MAPK and others). The Latin words (such as in vivo) should be written in cursive. Also use uniform style for titles and subtitles (f. e. 2.1. Secreted mucins, 3.1. Mucins and Biliary Tract Development). The word fetal-foetal should be written in a uniform way throughout the whole manuscript. The table is in this form not very transparent – maybe the vertical lines in the table could make it more clear.

In summary, the manuscript is very interesting and could have an impact on scientific community. After minor revisions I would recommend this manuscript to be accepted to the International Journal of Molecular Sciences.

Author Response

Thank you very much for your favourable review and time spent on reviewing the manuscript. As per suggestion, the manuscript was thoroughly revised, and all shortcomings have been corrected. Our work originally does not contain different fonts, changes were made after converting the original text by the editorial system. Similarly, Table 1 also previously contained vertical lines. It seems that such a table layout is automatically changed by the editing system. We have restored the original look of the table, but we have no influence on its final appearance. We added the explanation of abbreviations used for the first time in the text, although all abbreviations are also explained in a separate subchapter. The missing words were added, and mistakes were corrected. All Latin words are now written in italics (cursive).

Reviewer 2 Report

This paper reviewed the role of Mucins fo hepatobiliary caricinogenesis. Anyway, there area possibiltiy of relationship Mucins and biliary differentiation rather than hepatobiliary carcinogesis. Thus, I would be better to discuss the reasons regarding that Mucins are related with carcinogenesis rather than biliary differentation.

# Minor comment - In manucript, size and style of text is uneven, please recheck the style and size of text.

Author Response

Thank you very much for a review and time spent on reviewing the manuscript. If we have understood the suggestion of the Reviewer well, asking us to discuss the correlations between „mucins and carcinogenesis rather than biliary differentiation”, then we need to not that we have indeed aimed to present the results of available literature, considering both in vitro and animal model studies, as well as clinical research on patients with different types of primary liver cancers (PLC). These results showed such correlation (chapter 4, Figure 1, Table 1). However, determination of actual mechanisms/causes of those interactions is either very difficult or only possible for some mucin types (e.g. MUC1 as a biliary epithelial marker). The general correlations are confirmed by all of the above-mentioned models of research, as well as clinical-pathological data and, in case of some mucins (e.g. MUC16), genetic studies (subchapter 4.7).

In the course of gastrointestinal (GI) tract carcinogenesis (including hepatobiliary carcinogenesis), quantitative changes of mucin amount are observed (e.g. MUC1 overexpression). Additionally, qualitative differences in mucin molecule structure (e.g. disturbances of glycosylation), changes in cellular localisation (cytoplasmic or mixed expression, instead of membrane expression), as well as intensification of oncogenic functions described in detail in other cancers of the GI tract (ref. 16,29,30,31). Mucin changes appear with progression of histological and clinical changes in the case of hepatobiliary tract cancers (BTC), which was described in sub-chapters of this manuscript (e.g. 4.3.1., 4.3.2, 4.4., 4.5, 4.6, 4.7). The order of changes is still hard to determine (cause, effect). The knowledge about differential mucin expression on different types of BT carcinogenesis is still incomplete, with the current research serving to explain these dependencies.

Description of the role of mucins in the hepatobiliary tract (HBT) carcinogenesis, included in chapter 3 (3.1 and 3.2), aimed to present the possibility of changes in activity of the mucin genes, which play a role in physiological HBT differentiation, as well as are responsible for phenotypic properties of cholangiocytes/hepatocytes, which in turn are essential in hepatobiliary carcinogenesis. The exact mechanisms of control of these genes’ expression in hepatobiliary tract differentiation are still intensively researched (ref. 71-73).

We have rechecked the style and size of all our text. We would like to add that our work originally did not contained different fonts, changes were made after converting the original text by the editorial system. We are sorry for that fact.